# LPS-Induced Liver Injury of Magang Geese through Toll-like Receptor and MAPK Signaling Pathway

**DOI:** 10.3390/ani13010127

**Published:** 2022-12-28

**Authors:** Bingxin Li, Longsheng Hong, Yindan Luo, Bingqi Zhang, Ziyu Yu, Wanyan Li, Nan Cao, Yunmao Huang, Danning Xu, Yugu Li, Yunbo Tian

**Affiliations:** 1College of Animal Science & Technology, Zhongkai University of Agriculture and Engineering, Guangzhou 510225, China; 2Guangdong Province Key Laboratory of Waterfowl Healthy Breeding, Guangzhou 510225, China; 3College of Veterinary Medicine, South China Agricultural University, Guangzhou 510642, China

**Keywords:** LPS, liver injury, goose, Toll-like receptor, MAPK

## Abstract

**Simple Summary:**

LPS is one of the main virulence factors of Gram-negative bacteria. In the breeding process of geese, due to the influence of environmental factors, harmful bacteria easily breed in the water body, resulting in an increase in the concentration of LPS in the goose. The liver has an important function of clearing LPS, but excess LPS can induce liver damage, resulting in a reduced production performance for the goose. In our study, we found that LPS induces inflammatory damage in the liver. Further, through transcriptome sequencing analysis, we screened 727 differentially expressed genes for LPS-induced liver injury, and we performed enrichment analysis. The results showed that LPS-induced liver injury in geese may be the result of the joint action of Toll-like receptor, MAPK, NOD-like receptor, FoxO, and PPAR signaling pathway. Among them, the TLR7-mediated MAPK signaling pathway plays a major role.

**Abstract:**

Lipopolysaccharide (LPS) is one of the main virulence factors of Gram-negative bacteria. In the process of waterfowl breeding, an inflammatory reaction due to LPS infection is easily produced, which leads to a decline in waterfowl performance. The liver plays a vital role in the immune response and the removal of toxic components. Therefore, it is necessary to study the mechanism of liver injury induced by LPS in goose. In this study, a total of 100 1-day-old goslings were randomly divided into a control group and LPS group after 3 days of pre-feeding. On days 21, 23, and 25 of the formal experiment, the control group was intraperitoneally injected with 0.5 mL normal saline, and the LPS group was intraperitoneally injected with LPS 2 mg/(kg body weight) once a day. On day 25 of the experiment, liver samples were collected 3 h after the injection of saline and LPS. The results of histopathology and biochemical indexes showed that the livers of the LPS group had liver morphological structure destruction and inflammatory cell infiltration, and the levels of ALT and AST were increased. Next, RNA sequencing analysis was used to determine the abundances and characteristics of the transcripts, as well as the associated somatic mutations and alternative splicing. We screened 727 differentially expressed genes (DEGs) with *p* < 0.05 and |log_2_(Fold Change)| ≥ 1, as the thresholds; GO and KEGG enrichment analysis showed that LPS-induced liver injury may be involved in the Toll-like receptor signaling pathway, MAPK signaling pathway, NOD-like receptor signaling pathway, FoxO, and PPAR signaling pathway. Finally, we intersected the genes enriched in the key pathway of LPS-induced liver injury with the top 50 key genes in protein–protein interaction networks to obtain 28 more critical genes. Among them, 17 genes were enriched in Toll-like signaling pathway and MAPK signaling pathway. Therefore, these results suggest that LPS-induced liver injury in geese may be the result of the joint action of Toll-like receptor, MAPK, NOD-like receptor, FoxO, and PPAR signaling pathway. Among them, the TLR7-mediated MAPK signaling pathway plays a major role.

## 1. Introduction

In 2021, the world’s meat goose production was about 660 million geese, among which Asia accounts for 96%. China is the central region of goose raising, and it is also the world’s largest meat goose production country [1]. Magang goose enjoys a high reputation in Guangdong Province for its excellent growth performance and meat quality, and its breeding scale is the largest, accounting for about 80% of the total number of geese raised in the province [2]. In the process of goose breeding, due to the influence of environmental factors, such as temperature and breeding scale, more harmful bacteria easily breed in water, which leads to an increase in lipopolysaccharide (LPS) concentration in goose blood, reducing the reproductive performance of geese and the quality of the goslings [3]. Gram-negative bacteria produce the toxic substance LPS, which can seriously damage the normal physiological activities of mammals and birds, significantly reduce the feed intake, body weight, daily egg production, and egg weights of laying hens, and reduce body immunity and even lead to the death of the geese or goose embryos [4,5,6]. LPS is a powerful activator of innate immune responses. After LPS enters the body, it stimulates an innate immune response, inducing cells to produce and release a series of inflammatory mediators, resulting in the rapid spread of toxicity in the body and accelerating the process of inflammation [7,8].

Due to its anatomical location, the liver is constantly exposed to a variety of antigens from the gut, including bacteria and/or their toxic products, such as LPS [9]. Therefore, the liver not only plays a crucial role in metabolizing immune responses, fighting bacterial/viral infections, and removing toxic components, but it also has a variety of functions, such as synthesis, storage, decomposition, and excretion, and regulating the balance of the body’s internal environment [10,11]. Moreover, different cell groups in the liver can be involved in adjusting the balance between innate immunity and adaptive immunity, which is an important natural barrier for the body to defend against aggression [12,13]. Immune cells in the liver can eliminate antigens from the gut via powerful phagocytosis and the secretion of a variety of cytokines and chemokines, and inactivate antigens in the presence of bacteria or their metabolites to avoid the imbalance of innate immunity and adaptive immunity caused by a large number of antigenic substances [9]. Increased LPS levels in the liver stimulate hepatic macrophages, leading to the release of inflammatory factors, such as interleukin-1, interleukin-6, and tumor necrosis factor-α, leading to liver inflammation and liver injury. [14,15]. In patients with acute liver failure, persistent infection also leads to a decrease in the liver’s ability to clear LPS, a 2- to 7-fold increase in the release of inflammatory cytokines tumor necrosis factor-α (TNF-α), interleukin-1β (IL-1β), and interleukin 6 (IL6), and an increase in circulating aminotransferases, thus aggravating liver function injury [16]. In addition, LPS activates macrophages to produce TNF-α and other potential cytotoxic pro-inflammatory mediators, it can increase the neurological complications caused by liver disease, and it may be involved in the pathogenesis of hepatic encephalopathy [17,18]. LPS-induced liver injury will further lead to a decline in all aspects of goose production performance. However, the underlying mechanism of LPS-induced liver injury in geese remains poorly understood.

In this study, 100 Magang geese were randomly divided into a control group and an LPS group. After different treatments, the livers of the two groups of Magang geese were examined for biochemical indicators, histopathology, and RNA sequencing (RNA-seq) analysis to grasp the effects of LPS on goose livers and the potential mechanisms, which provide a new theoretical basis for the prevention and treatment of LPS-induced liver in-jury in goose.

## 2. Materials and Methods

### 2.1. Ethics Statement

The experiments involved in this study are in full compliance with the relevant specifications formulated by the Ministry of Agriculture of China. For all geese (Anser cygnoides), the care, slaughter, and experimental procedures were evaluated and approved by the Animal Ethics Committee of Zhongkai University of Agriculture and Engineering (Approval code: NO. 202112-01).

### 2.2. Animals and Sample Preparation

One hundred 1-day-old Magang geese, half of them male and half of them female, were purchased from Qingyuan Shixing Biotechnology Co., LTD (Qingyuan, China). The goslings were fed freely (including regular feed and vegetables) and water from 1 day of age. The daily light environment is 12/12 h light/dark cycle. After 3 days of prefeeding, the geese were randomly divided into two groups, with 5 replicates per group and 10 geese per replicate. There was no significant difference in body weight between each replicate. On days 21, 23, and 25 of the formal experiment, the control group was intraperitoneally injected with 0.5 mL normal saline, and the LPS group was intraperitoneally injected with LPS (Sigma, St. Louis, MO, USA) (2 mg/kg body weight) once a day. On day 25 of the experiment, 3 h after injection of saline and LPS, one goose was randomly selected from each replicate in two groups for carbon dioxide anesthesia and painless death for liver collection (*n* = 5). All geese were fasted for 12 h before sampling. The obtained liver samples were partially snap-frozen in liquid nitrogen and stored in a refrigerator at −80 °C, and partially stored in 4% paraformaldehyde (PFA, BL539A, Biosharp, Guangzhou, China).

### 2.3. Histopathological Examination of the Liver

The livers of the geese were fixed in 4% paraformaldehyde for 3 days, then the livers were dehydrated, embedded in paraffin, and cut into 5 μm before hematoxylin and eosin (H.E.) staining (*n* = 3). Finally, the changes in liver structure and cellular substructure were observed and photographed using a light microscope (OLYMPUS, Tokyo, Japan).

### 2.4. Liver Biochemical Index Detection

The liver sample was weighed, and pre-cooled normal saline was added according to the ratio of weight (g): volume (mL) = 1:9 (*n* = 5). The liver was then fully ground with a sample breaker and centrifuged at 3000 rpm for 10 min to collect the supernatant. Subsequently, the levels of alanine transaminase (ALT, Nanjing Jiancheng Corp., Nanjing, China) and aspartate aminotransferase (AST, Nanjing Jiancheng Corp., Nanjing, China) were measured using the corresponding kits. The results were statistically analyzed and mapped using GraphPad Prism 7.00 software (Prism, San Diego, CA, USA). The differences between groups were analyzed using Student’s t-tests, and the values are expressed as means ± standard deviation (SD). * indicated *p* < 0.05, and ** indicated *p* < 0.01.

### 2.5. RNA Extraction, Library Construction, and Sequencing

In order to explore the mechanism of LPS-induced liver injury, we constructed six cDNA libraries (three biological replicates in each control group and LPS group). First, total RNA was extracted, isolated, and purified with Trizol reagent (Invitrogen, Carlsbad, CA, USA). A NanoDrop ND1000 (NanoDrop, Wilmington, DE, USA) and NanoDrop ND1000 (NanoDrop, Wilmington, DE, USA) were used to conduct quality control on the amount and purity of total RNA, and the integrity of the RNA was tested. We selected total RNA with a concentration > 50 ng/μL and a RIN value > 7.0, OD260/280 > 1.8, detected via agarose gel electrophoresis integrity, and then we purified mRNA from the total RNA (5 ug) using Dynabeads Oligo (DT) (Thermo Fisher, San Jose, CA, USA). Two rounds of purification were followed by fragmentation using the Magnesium RNA Fragmentation Module (NEB, CAT.E6150, Ipswich, MA, USA) at high temperature. The RNA fragments were then reverse transcribed using SuperScript™ II Reverse Transcriptase (Invitrogen, Cat. Carlsbad, CA, USA) to generate cDNA. Then, we used the synthesized U-labeled second-stranded DNAs with *E. coli* DNA polymerase I (NEB, the m0209, Ipswich, MA, USA), RNase H (NEB, Cat.m0297, Ipswich, MA, USA), and dUTP solution (Thermo Fisher, CAT.R0133, San Jose, CA, USA) to form a library with a fragment size of 300 bp ± 50 bp. Finally, these cDNAs were sent to LC-Bio Biotechnology, Inc. Six gene libraries were obtained via double-ended sequencing (sequencing mode PE150) using Illumina Novaseq™ 6000 (LC Bio Technology CO., Ltd. Hangzhou, China).

### 2.6. Sequencing Analysis

After sequencing, the raw data were filtered using Cutadapt (https://cutadapt.readthedocs.io/en/stable/, version:cutadapt-1.9, accessed on 6 May 2022) to obtain clean readings with the removal of the adapters, polyA and polyG, more than 5% of unknown nucleotides (N), and more than 20% of low-quality (Q-value ≤ 20) bases [19]. Then, FastQC (http://www.bioinformatics.babraham.ac.uk/projects/fastqc/, 0.11.9, accessed on 11 May 2022) was used to verify and to obtain high-quality cleaning data (including Q20, Q30, and the GC contents of clean data) [20]. The high-quality cleaning data were mapped to the Anas cygnoides genome sequence (https://www.ncbi.nlm.nih.gov/genome/?term=Anser+cygnoides, accessed on 12 May 2022), and also based on the Anas cygnoides genome, using the HISAT2 (https://daehwankimlab.github.io/hisat2/, version:hisat2-2.0.4, accessed on 25 May 2022) package map with the Anas cygnoides genome [21].

### 2.7. Analysis of SNP/InDel and the Prediction of Alternative Splices (AS)

According to the HISAT2 comparison results between reads and the Ac genomic sequences, Samtools (0.1.19) was used to call the variants of transcripts, and ANNOVAR was used to annotate single nucleotide variants (SNVs) and insertion–deletion (INDEL) and to analyze the functions, genomic loci, and variant types of SNVs [22,23]. In addition, we used the comparison results to identify alternative splicing events via rMATS (version 4.1.1) (http://rnaseq-mats.sourceforge.net, accessed on 28 May 2022) and analyzed the differences between samples. We identified AS events with a false discovery rate (FDR) of <0.05 in a comparison as significant AS events. The categories of alternative splicing are SE (skip exon), A3SS (substitute for 3′ splicing site), A5SS (substitute for 5′ splicing site), MXE (mutually exclusive exon), and RI (retain intron).

### 2.8. Analysis of Differentially Expressed Genes

All transcriptomes from all samples were merged to reconstruct a comprehensive transcriptome using StringTie (http://ccb.jhu.edu/software/stringtie/, version:stringtie-1.3.4d, accessed on 2 June 2022) and gffcompare software (http://ccb.jhu.edu/software/stringtie/gffcompare.shtml, version:gffcompare-0.9.8, accessed on 2 June 2022) [21,24]. The expression levels of all transcripts were then estimated using StringTie and ballgown (http://www.bioconductor.org/packages/release/bioc/html/ballgown.html, accessed on 2 June 2022), and the mRNA expression levels were estimated by calculating the FPKM (fragment per kilobase of transcript per million mapped reads) value. DESeq2 software was used to analyze the gene differential expression between the control group and LPS group; *p* < 0.05, fold change (FC) > 2 or FC < 0.5 were considered as differentially expressed genes (DEGs) [25].

### 2.9. GO Enrichment and KEGG Pathway Enrichment Analysis

DEGs performed Gene Ontology (GO) and KEGG functional enrichment analysis using the Goseq R software package and the Kyoto Encyclopedia of Genes and Genomes (KEGG) enrichment analysis using the KOBAS online tool (http://kobas.cbi.pku.edu.cn/genelist/, accessed on 13 July 2022), respectively. A value of *p* < 0.05 represented a significant difference. Finally, we visualized the enrichment analysis results of GO (top 13 biological processes, 6 cellular components, and 5 molecular functions) and KEGG (top 25), based on the *p*-values. GO enrichment analysis circle plot is plotted by the online site ChiPlot (https://www.chiplot.online/, accessed on 13 July 2022). The outermost layer of the GO enrichment analysis circle plot represents the enriched GO ID. The second layer represents the number of background genes in the pathway, and the depth of the color represents the size of the *p* value. The third layer represents the Rich factor, with red representing the biological process, light green representing the cellular component, and earthier yellow representing the molecular function. The KEGG pathways plot was created using the R package ggplot2 (https://ggplot2.tidyverse.org, accessed on 14 July 2022).

### 2.10. Construction of a Gene Network

We selected significant KEGG pathways related to LPS-induced liver injury, including the cytokine–cytokine receptor interaction, Toll-like receptor signaling pathway, NOD-like receptor signaling pathway, MAPK signaling pathway, FoxO signaling pathway, PPAR signaling pathway, p53 signaling pathway and Apoptosis. Then, the genes enriched in the pathways were imported into the String database (V 11.5) [26]. The minimum comprehensive score of the key gene network was set as 0.4, and the nodes interrupted in the network were hidden. The obtained analysis was further imported into Cytoscape software (V 3.9.1) for a better visualization and analysis of the gene networks [27]. In this gene network, each node represents a biomolecule, the edges represent interactions between nodes, and the sizes of the nodes are adjusted according to the number of edges linked to other genes (node degree): the larger the node degree, the larger the node.

### 2.11. RNA-seq Validation via qRT-PCR

To verify the reliability of RNA-seq results, 10 representative genes of the Toll-like receptor signaling pathway and MAPK signaling pathway were selected for qRT-PCR (*n* = 3). RNA with an A260/280 ratio of between 1.8 and 2.1 and an A260/230 ratio of >2.0 was reverse transcribed to synthesize cDNA using the TaKaRa reverse transcription kit (RR036A, TaKaRa, Beijing, China). Then, the SYBR Premix Ex TaqTM ii (TliRNaseH Plus) kit was used for real-time fluorescence quantification using an ABI QuantStudio7 Q7 real-time PCR instrument. *ACTB* was used as the reference gene, and the primer sequences are shown in Appendix A. Each reaction mixture contained 5 μL PowerllP SYBR Green Master Mix (2×), 0.5 μL forward primers, 0.5 μL reverse primers, 0.5 μL cDNA, and 3.5 μL ddH_2_O. The reaction process of qRT-PCR was followed by 40 cycles of 50 °C for 2 min, 95 °C for 2 min, then 95 °C for 15 s and 55 °C for 1 min. The relative mRNA expression levels of each gene were analyzed using the 2^−ΔΔCT^ method, and the results were statistically analyzed and mapped using GraphPad Prism 7.00 software (Prism, San Diego, CA, USA).

## 3. Results

### 3.1. Effect of LPS on the Liver Histopathology of Magang Geese

The histological observation in Figure 1 shows that the livers of the control group were normal, with intact lobules, neatly arranged liver cords, and clear nucleoli. In the LPS group, the liver morphology structures were destroyed, hepatocyte swelling was obvious, cytoplasmic content was absent, and liver cord structures were disordered, the hepatic sinusoid structures could not be clearly observed, and many inflammatory cells were infiltrated. Comparing the histological observation of the two groups, we found that LPS significantly induced the destruction of liver morphology and structure and led to liver inflammation.

### 3.2. Effects of LPS on the Liver Biochemical Indices of Goose

In two groups, the levels of ALT and AST in the livers of geese were measured, and the results are shown in Figure 2. The levels of AST in the livers of LPS-induced goslings were significantly higher than those of the control group (*p* < 0.01), while the levels of ALT were not significantly different but showed a tendency to increase under the induction of LPS. ALT/AST is also an important index for evaluating liver function. LPS-induced ALT/AST in gosling livers were significantly lower than those in the control group (*p* < 0.01).

### 3.3. Overview of RNA Sequencing in Magang Goose Livers

In this study, six RNA-seq libraries were constructed to comprehensively analyze the related genes involved in LPS-induced liver injury in Magang geese. A total of 292.4 million raw reads were identified in the RNA-seq library, with an average of 48.7 million raw reads per library. After quality control, the transcriptomes of liver yielded approximately 281 million (46,792,734 on average) clean reads, and the valid ratio ranged from 95.65% to 96.22%. As shown in Table 1, the lowest GC content (the proportion of GC content) and Q30 (the proportion of bases with mass value ≥ 30) of the samples were 47.5% and 96.98%, respectively. Additionally, more than 37,547,059 high-quality reads were mapped to the Aner cygnoides domesticus (Ac) genome for gene expression analysis. Unique mapped reads represent the number of reads that can only be mapped to one location in the genome, while multi mapped reads are the number of reads that can be mapped to multiple locations in the genome. The comparison found that uniq mapped reads and multiple map reads accounted for 84.12–85.26% and 1.41–1.79%, respectively. On average, 80.70% reads were mapped to the number of sense strands on the genome, and 42.27% reads were mapped to the number of anti-sense strands on the genome (Appendix A).

### 3.4. Annotation and Classification of SNV/InDel and the Prediction of Alternative Splicing

As shown in Table 2, 428,911 to 487,581 SNVs were detected in the livers of the control and LPS groups. The proportion of transitional SNVs in all SNV sites was the largest, accounting for 72.71–72.84%. The most common variation was C-T, followed by A-G and C-G. Of the SNV and InDels mainly distributed in “intron”, “intergenic”, and “UTR3” (Figure 3A,B). Furthermore, through sequence alignment with the Anas cygnoides genome, we obtained the chromosomal positions of each transcript and detected five different splicing patterns in Magang goose livers, namely SE (skipped exon), A3SS (alternative 3′ splice site), A5SS (alternative 5′ splice site), MXE (mutually exclusive exons), and RI (retained intron) (Figure 3C). The number of differentially expressed AS events was 1304, and the predicted alternative splices is mainly concentrated in skipped exon, which is over 800.

### 3.5. Identification of Differentially Expressed Genes

To identify potential candidate genes for LPS-induced liver injury, the expression levels of genes in the livers of control and LPS groups were measured. *p* < 0.05 and |log_2_(Fold Change)| ≥ 1 were set as the standard for differential expression. DEGs were clustered using the pheatmap R package (https://CRAN.R-project.org/package=pheatmap, v1.0.12, accessed on 14 July 2022) according to their fold change (Figure 4A). Additionally, a scatter plot was produced using the R package ggplot2 (https://ggplot2.tidyverse.org, accessed on 14 July 2022), according to the *p* value and FPKM (Figure 4B). In the scatter plot, blue dots indicated downregulated genes, red dots indicated upregulated genes, gray dots indicated non-differently-expressed genes, and the sizes of the dots indicated the significance of the genes. The larger the dots, the stronger the significance. Based on Figure 4, the DEGs for three individuals in each group had good reproducibility and a total of 727 genes were differently expressed between liver in the control group and the LPS group, including 424 upregulated genes and 303 downregulated genes (Appendix A).

### 3.6. Analysis of GO Annotation and KEGG Pathway

To gain valuable insight into the molecular functions of the genes potentially associated with LPS-induced liver injury, the identified DEGs were analyzed for enrichment according to GO. Here, we showed the top 13 most significant biological processes, the top six most significant cellular components, and the top five most significant molecular functions (Figure 5A and Table 3). We found that most DEGs were involved in inflammation, such as the Toll-like receptor 2 signaling pathway, the defense response against bacteria, the regulation of the apoptosis process, and the immune response, suggesting that LPS may induce liver injury mainly through the inflammation and immune responses.

To further explore the pathways through which LPS induces liver injury, the KOBAS online tool was used for KEGG analysis. The top 25 enriched KEGG pathways are shown in Figure 5B. The most enriched significantly pathways are the cytokine–cytokine receptor interaction, Toll-like receptor signaling pathway, NOD-like receptor signaling pathway, and the MAPK signaling pathway. All of these pathways are involved in inflammation. In addition, DEGs were also significantly enriched in the FoxO signaling pathway, PPAR signaling pathway, p53 signaling pathway, endocytosis, and apoptosis, which are involved in the body’s inflammatory and immune responses.

### 3.7. Analysis of Interaction Network Analysis

To further explore the key DEGs, we performed an interaction network analysis of the DEGs. According to the STRING database analysis, an interaction network consisting of 271 nodes and 762 edges was constructed using Cytoscape (Figure 6A). The size of dots indicates the degree of nodes, red indicates upregulated genes, and blue indicates downregulated genes. Subsequently, we selected the top 50 key genes based on the index of the degree (Figure 6B), which were at the key positions in the network and might be more critical than other genes. We found that *IL6* has the highest degree of these and may play an important role in LPS-induced liver injury. Finally, the intersection of 50 key genes with genes enriched in key pathways in KEGG analysis, such as cytokine–cytokine receptor interaction, Toll-like receptor signaling pathway, NOD-like receptor signaling pathway, MAPK signaling pathway, etc., yielded 28 key genes (Figure 6C and Table 4). Among the 28 genes, most of them were found to be in the Toll-like receptor signaling pathway (Appendix A) and the MAPK signaling pathway (Appendix A).

### 3.8. Validation of DEGs Using qRT-PCR

To further verify the accuracy of RNA-seq data, 10 DEGs that were present in 28 key DEGs and expressed in the Toll-like receptor signaling pathway and MAPK signaling pathway were verified via qRT-PCR. The results showed that the expression trend of RNA in qRT-PCR was consistent with that in RNA-seq, and there were significant differences (Figure 7). Therefore, the results indicated that LPS may induce liver injury in Magang geese through the Toll-like receptor signaling pathway and MAPK signaling pathway.

## 4. Discussion

In the process of goose breeding, due to the influence of environmental factors, such as temperature and breeding scale, more harmful bacteria, such as *E. coli* and *Salmonella* spp., are easily bred in water, which leads to an increase in LPS concentration in goose blood [3]. LPS can stimulate hepatic macrophages (Kupffer cells) to produce various pro-inflammatory cytokines, and then induce acute infection or inflammation [28], reducing the reproductive performances of geese and the quality of the goslings, as well as the feed intakes and growth rates of growing young geese [29]. In addition, LPS binding proteins are mainly synthesized by hepatocytes. When LPS enters the body, LPS reacts with LBP in the liver, forming an LPS–LBP complex to promote the recruitment of immune cells and the release of inflammatory factors. If the LPS source cannot be removed, it will further lead to septic shock and multiple organ failure [30].

The histomorphological structure of the liver reflects the healthy state of the liver. In this study, compared with the control group, the morphologies of liver in the LPS group were destroyed, hepatocyte swelling was obvious, cytoplasmic content was missing, liver cord structures were disordered, hepatic sinusoidal structure could not be clearly observed, and a large number of inflammatory cells were infiltrated. A similar phenomenon was observed in previous studies of LPS-induced liver injury, which induces hepatocyte swelling, the loss of liver cord structure, the infiltration of inflammatory cells, and the exudation of red blood cells in mice [31]. In addition, ALT and AST levels are important indicators for evaluating liver injury [32]. In this study, the levels of AST and ALT in the liver of geese induced by LPS were increased, and ALT/AST were significantly downregulated. Therefore, the histomorphological structures and biochemical criteria suggested that the model of goose liver injury induced by LPS was successfully established.

Based on successful modeling, we further studied the molecular mechanism of LPS-induced liver injury via RNA-seq analysis. There were 43,177,160–49,660,302 genes expressed in the liver transcriptome of Magang geese, and an average of 40,386,605 high-quality reads per sample were mapped to the Anas cygnoides genome. RNA-seq measures structural variation, such as fusion transcription or mutation, along with gene expression. Somatic single nucleotide variation (SNV) is the simplest type of mutation, involving only a single base variation [33]. The investigation of SNV may provide an insight into its effect on gene/protein loss and disease risk [34]. InDels are a class of major genomic variants involving the addition or loss of one or more nucleotides of a DNA sequence throughout the genome, mainly detected from DNA-seq data [35,36]. There were 456,898 to 487,581 SNVs in the livers of the control and LPS-induced groups. The most common change was C-T, followed by A-G and C-G. The annotations of SNV and InDel were mainly distributed in “intron”, “intergenic”, and “UTR3”, indicating that the liver SNPs and InDels of LPS-induced Magang goose were mainly in “intron”, “intergenic”, and “UTR3”. Further, the discovery of these SNVS and InDels may improve our understanding of LPS-induced liver injury. Additionally, we found that the predicted number of variable splices in each sample was mainly concentrated in SE, which indicated that the main AS in the process of LPS-induced liver injury was SE. At present, there are few studies on SNV/INDEL and AS in geese. The screening analysis and annotation of SNP/INDL and AS in this study improved our understanding of the potential biology of LPS-induced liver injury in geese. LPS-induced liver injury can lead to changes in the expression levels of related genes, and further study of DEGs can understand the regulatory mechanisms of LPS-induced liver injury. In this study, we found a total of 727 DEGs in the LPS-induced liver injury group and the control group, and these genes were enriched using GO and KEGG analysis. GO analysis showed that many DEGs were significantly enriched in GO terms related to inflammatory and immune responses, and the cellular response to LPS, such as the Toll-like receptor 2 signaling pathway, acute-phase response, the defense response to bacteria, and chemokine activity. Further, KEGG enrichment analysis confirmed that DEGs were more focused and involved in the Toll-like receptor signaling pathway, MAPK signaling pathway, NOD-like receptor signaling pathway, FoxO signaling pathway, and the PPAR signaling pathway. It is well known that Toll-like receptors (TLRs) are pathogen recognition receptors that coordinate innate and subsequent adaptive immune responses. On the one hand, its activation can protect the body by stimulating the innate immune response and enhancing the acquired immune response, and the persistent inflammatory reaction caused by it will also cause damage to the body [37,38]. The MAPK signaling pathway plays an important role in the production of pro-inflammatory mediators in TLR signaling [39]. In addition, as a key regulator of immune response, NLR mutations may induce inflammation by interfering with the NF-κB, MAPK, and/or caspase-1 signaling pathways [40]. FoxO also plays an important role in immune responses and inflammation, and it has been shown that FoxO3 can regulate LPS-activated liver inflammation by reducing pro-inflammatory factors [41]. PPARα and PPARγ reduce inflammatory damage by downregulating JAK-STAT, AP-1, and NF-κB signaling pathways [42,43]. The accumulation of inflammation can further lead to liver damage [44]. These results suggested that LPS-induced liver injury in geese may be the result of the joint action of Toll-like receptor, MAPK, NOD-like receptor, FoxO, and PPAR signaling pathway.

To further explore the role of key genes in LPS-induced liver injury in Magang geese, we intersected the genes enriched in the key pathway of LPS-induced liver injury with the top 50 key genes in PPI to obtain 28 more critical genes. In addition, we found it interesting that most of the 28 genes (17 genes) were enriched in the Toll-like signaling pathway and MAPK signaling pathway. Therefore, we selected 10 genes out of the 17 genes, and the gene expression via qRT-PCR was consistent with the RNA sequencing results, which further verified the reliability of the above analysis. The Toll-like receptor signaling pathway is not activated in normal liver, but it can be activated by hepatocytes in a pathological state [45]. Relevant studies have shown that hepatocytes can express TLR1-9 and MyD88 transcripts and respond to a variety of pathogen-associated molecular patterns (PAMP), participating in the uptake and elimination of LPS [46,47]. In this study, we found that the expression of TLR7 was significantly upregulated during LPS-induced liver injury. TLR7 belongs to a sub-family of TLRs (including TLR3, TLR7, TLR8, and TLR9), which exclusively localize to intracellular compartments and possess a potent ability to induce type I IFN (interferon) [48]. It has been found that TLR7 is significantly upregulated in monocyte-derived DCs (MODC) of selected donors after LPS stimulation, inducing MODC to produce a variety of pro-inflammatory and anti-inflammatory cytokines [49]. Additionally, it has been found that inflammatory factors produced by inflammatory stimuli can upregulate TLR7 expression in Hep3B hepatocytes, and that pro-inflammatory stimuli activate TLR7 transcription through the NF-κB binding motif in this region, which can be blocked via the NF-κB binding site mutation or the addition of NF-κB inhibitors [50]. Many studies have also shown that viral infection and chronic inflammation induce TLR7 expression in a variety of non-native TLR7-expressing cell types [50,51,52]. In addition, TLR7 can also activate IFR7 and induce the expression of type I interferon (IFN-α/β) [53,54]. Moreover, we found that *MAP3K8*, *NFKBIA*, *JUN*, *IL6*, and *NFKB2* were significantly differentially expressed in this study, and all of them were upregulated in the process of LPS-induced liver injury. It is generally believed that TLR7 induces the expression of inflammatory factors through the MyD88-TRAF6-TAK1-MAPK/IKK-AP-1/NF-κB pathway [53,55,56]. Additionally, the TLRs receptor-induced activation of MAP3K8 can positively regulate the MAPK signaling pathway in inflammation [57].

MAP3K8, also known as Tumor Progression Locus 2 (TPL2) or COT, is a MAP3K that is downstream of the IL1R and TLR receptors. It is a major mediator of liver inflammation [57,58]. In our study, we found that the TLRs-like receptor signaling pathway and the MAPK signaling pathway intersect at MAP3K8. Studies have found that MAP3K8 can regulate NF-κB and Activator protein-1 (AP-1) to regulate the production of inflammatory factors [59]. LPS can induce the MAPK pathway to activate AP-1 and NF-κB transcription factors through the Toll-like receptor pathway, thereby promoting the activation of pro-inflammatory genes, leading to liver inflammation [60]. AP-1 is formed by the interaction of JUN and FOS families to form homologous pairs or heterodimers; these are mandatory transcription factors in inflammation and innate immunity [61]. NF-κB plays an important role in immunity, inflammation, stress, and other aspects. Under the induction of the TLR receptor, NF-κB and AP-1 can stimulate the production of inflammatory factors, such as IL6, TNFα, etc. [62,63]. We found that IL6 was significantly upregulated in RNA sequencing and qRT-PCR results, and it played an important role in the protein–protein interaction network, suggesting that IL6 plays an important role in LPS-induced liver injury. IL6 has pro-inflammatory and anti-inflammatory properties, and it contributes to liver homeostasis by regulating metabolic function, regeneration, and anti-infection defense [64]. In acute inflammation, IL6 can rapidly induce the liver to produce a wide range of acute-phase proteins, such as C-reactive protein, antitrypsin, etc. C-reactive protein can activate the classical pathway of the complement cascade, and antitrypsin can inactivate proteases released by pathogens and dead cells [65,66]. However, excessive IL6 expression can promote monocyte accumulation at the site of injury through sustained MCP-1 secretion, vascular proliferation, and the anti-apoptotic function of T cells, further leading to liver inflammation and liver injury [67,68,69]. In addition, IL6 is an NF-κB target, and activating both NF-κB and STAT3 in non-immune cells triggers a positive feedback loop for NF-κB activation via the IL6-STAT3 axis. The synergistic interaction between NF-κB and STAT3 induces the hyperactivation of NF-κB, enhancing chemokine and IL6 expression, followed by the accumulation of immune cells, leading to inflammation and local homeostasis dysregulation [70,71]. Therefore, the up-regulation of IL6 expression is, on the one hand, a response to LPS-induced liver inflammation, and on the other hand, its overexpression can further lead to liver injury. These results suggest that the TLR7-mediated MAPK signaling pathway may play a major role in LPS-induced liver injury in geese, which may be related to the activation of NF-κB by TLR7 to promote the expression of IL6.5. 

In conclusion, this study found that LPS-induced liver injury in geese may be the result of the joint action of Toll-like receptor, MAPK, NOD-like receptor, FoxO, and PPAR signaling pathway. Among them, the TLR7-mediated MAPK signaling pathway plays a major role, which may be related to the overexpression of IL6. Although this study provides a valuable reference for understanding the LPS-induced liver injury of Magang goose, more in-depth studies are needed to validate these results.

## Figures and Tables

**Figure 1 animals-13-00127-f001:**
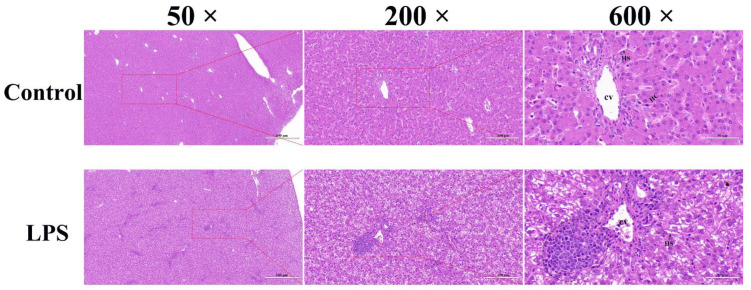
Histological observations of goose livers (50×, 200×, and 600×): CV: Central vein; HS: Hepatic sinusoid; HC: Hepatic cord; Oval area: Inflammatory cell infiltration.

**Figure 2 animals-13-00127-f002:**
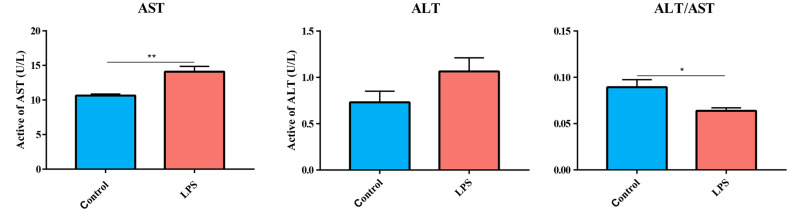
Effect of LPS-induced ALT and AST in the livers of goslings. * indicated *p* < 0.05, ** indicated *p* < 0.01.

**Figure 3 animals-13-00127-f003:**
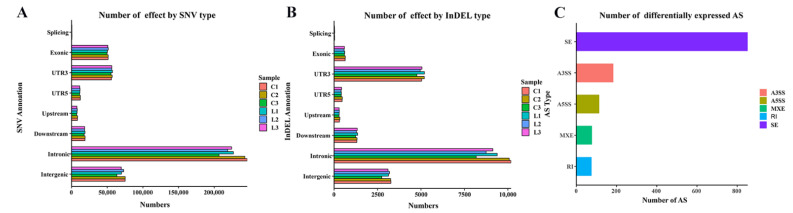
Annotation and classification of SNV/InDel and the prediction of alternative splicing. (**A**) The number of effects by SNV stype, (**B**) The number of effects by INDEL type. (**C**) The number of differentially expressed AS. SE stands for skipped exon, A3SS stands for alternative 3′ splice site, A5SS stands for alternative 5′ splice site, MXE stands for mutually exclusive exons, RI stands for retained intron.

**Figure 4 animals-13-00127-f004:**
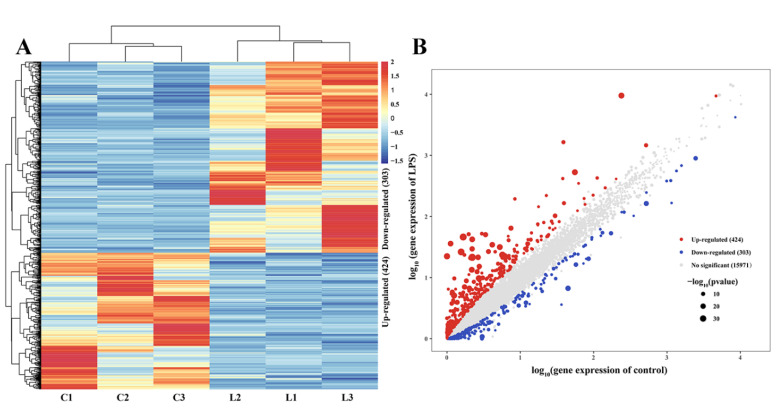
Differentially expressed genes in the livers of the control and LPS groups of goslings. (**A**) The heat map of control group vs. LPS group differentially expressed genes. (**B**) Gene expression distribution scatter plot. Blue dots indicated downregulated genes, red dots indicated upregulated genes, gray dots indicated non-differently expressed genes, and the sizes of the dots indicated the significance of the genes.

**Figure 5 animals-13-00127-f005:**
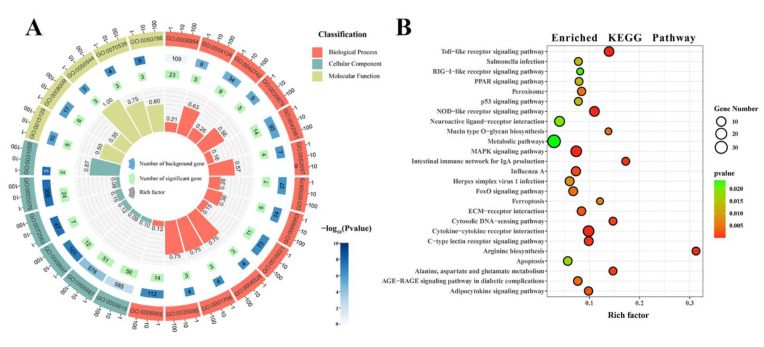
Analysis of GO and KEGG pathway enrichment of DEGs in goose liver. (**A**) GO enrichment analysis circle plot: The top 13 most significant biological processes, the top 6 most significant cellular components, and the top 5 most significant molecular functions. The outermost layer of the represents the enriched GO ID. The second layer represents the number of background genes in the pathway, and the depth of the color represents the size of the *p* value. The third layer represents the number of significant genes. The innermost columnar represents the rich factor, with red representing the biological process, light green representing the cellular component, and earthier yellow representing the molecular function. (**B**) The top 25 KEGG enrichment bubble plot: the size of the circle represents the number of genes, and the depth of the color represents the size of the *p* value.

**Figure 6 animals-13-00127-f006:**
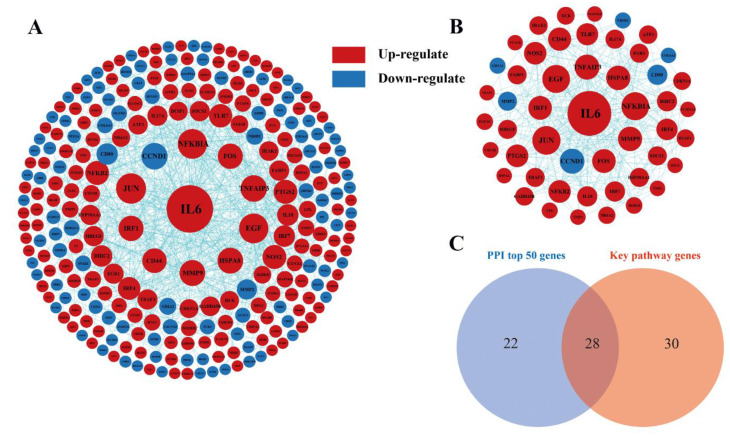
Interaction of differentially expressed gene protein network. (**A**) Protein–protein interaction network (PPI). (**B**) The network of 50 key genes. The red dot represents upregulation, the blue represents downregulation, and the size of the dot represents the size of the degree. The larger the dot, the stronger the significance. (**C**) The intersection of top 50 PPI genes with key pathway genes.

**Figure 7 animals-13-00127-f007:**
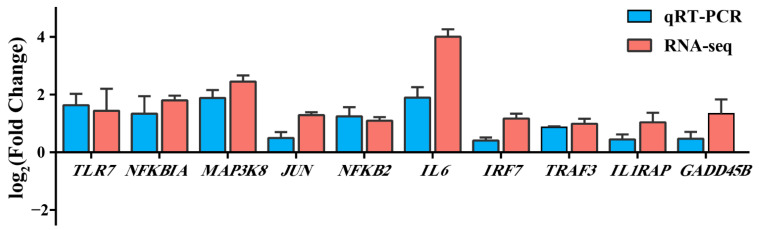
Validation of DEGs using qRT-PCR. The blue column represents qRT-PCR data, and the red column represents RNA-seq data.

**Table 1 animals-13-00127-t001:** Quality analysis of transcriptome sequencing.

Samples	Raw Data	Clean Data	Valid Ratio	GC Content	Q30 Value
C1	50,056,712	47,880,158	95.65%	48.00%	96.98%
C2	50,257,844	48,360,268	96.22%	47.50%	97.27%
C3	44,899,474	43,177,160	96.16%	48.50%	97.49%
L1	45,825,564	43,990,084	95.99%	48.50%	97.33%
L2	49,605,102	47,688,432	96.14%	48.50%	97.41%
L3	51,768,960	49,660,302	95.93%	48.50%	97.39%

**Table 2 animals-13-00127-t002:** SNV mutation types from goose livers.

Samples	A-G	C-T	A-C	A-T	C-G	G-T	Total	Transition	Transversion
C1	177,194	177,322	34,263	27,889	37,250	33,663	487,581	72.71%	27.29%
C2	176,721	176,522	34,305	28,051	37,260	34,097	486,956	72.54%	27.46%
C3	155,964	156,455	29,645	24,394	32,850	29,603	428,911	72.84%	27.16%
L1	167,120	167,508	32,409	26,307	35,285	32,242	460,871	72.61%	27.39%
L2	166,199	166,149	31,905	25,918	35,061	31,666	456,898	72.74%	27.26%
L3	165,059	166,306	32,207	26,134	35,100	32,109	456,915	72.52%	27.48%

**Table 3 animals-13-00127-t003:** GO ID associated with the GO term.

GO ID	GO Term	GO ID	GO Term
GO:0006954	Inflammatory response	GO:0006955	Immune response
GO:0034134	Toll-like receptor 2 signaling pathway	GO:0005615	Extracellular space
GO:0042742	Defense response to bacterium	GO:0050786	RAGE receptor binding
GO:0035879	Plasma membrane lactate transport	GO:0009897	External side of plasma membrane
GO:0042981	Regulation of apoptotic process	GO:0072562	Blood microparticle
GO:0052697	Xenobiotic glucuronidation	GO:0005576	Extracellular region
GO:0035095	Behavioral response to nicotine	GO:0031092	Platelet alpha granule membrane
GO:0006953	Acute-phase response	GO:0070538	Oleic acid binding
GO:0019221	Cytokine-mediated signaling pathway	GO:0008009	Chemokine activity
GO:0006526	Arginine biosynthetic process	GO:0050544	Arachidonic acid binding
GO:0001706	Endoderm formation	GO:0015129	Lactate transmembrane transporter activity
GO:0050679	Positive regulation of epithelial cell proliferation	GO:0005887	Integral component of plasma membrane

**Table 4 animals-13-00127-t004:** The 28 key genes in the intersection of top 50 PPI genes with key pathway genes.

Gene Symbol	Degree	Up/Down-Regulation
*IL6*	59	up
*JUN*	31	up
*EGF*	30	up
*NFKBIA*	30	up
*CCND1*	25	up
*TNFAIP3*	25	up
*HSPA8*	24	up
*FOS*	23	up
*NFKB2*	21	up
*NOS2*	21	up
*TLR7*	20	up
*IL18*	17	up
*CD80*	17	down
*BIRC2*	17	up
*IRF7*	17	up
*TRAF3*	15	up
*HSP90AA1*	15	up
*FABP1*	15	up
*GADD45B*	14	up
*DUSP1*	13	up
*CDKN1A*	13	down
*FGF10*	11	up
*IKBKE*	11	up
*CSF3R*	10	up
*LPL*	10	up
*MAP3K8*	9	up
*HSPA2*	9	up
*TRAF1*	9	up

## Data Availability

The data presented in this study are openly available in the Sequence Read Archive (SRA), PRJNA899715.

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
