# Peer review of "LPS-Induced Liver Injury of Magang Geese through Toll-like Receptor and MAPK Signaling Pathway"

_animals, 2022, doi:10.3390/ani13010127_

Round 1

Reviewer 1 Report

the article presented with professional style. for better understanding , the figure ligand may give more detail description.

the article showed that LPS may induce liver injury through the Toll-like receptor signaling pathway and the MAPK signaling pathway.this provided a valuable reference for under-standing the LPS-induced liver injury of Magang goose. the paper topic was suit to the journal, and the results encourage to be published , however  minor revision was needed..

Author Response

Thank you for your decision and constructive comments on my manuscript. We have carefully considered your suggestion and make some changes. We have tried our best to improve and made some changes in the manuscript, the detailed corrections are listed in the attachment.

Reviewer 2 Report

Research on LPS pathways is extremely important for the development of immunology. Not only because of the pathogenic effect of LPS, but also the possibility of using LPs for immunostimulation. However, an article in this form cannot be accepted for publication. The manuscript needs major revisions, especially the results section.

- Table S1 should contain the gene accession numbers.

- The introduction chapter contains information about the experiment procedure. this information should be limited to the material and methods section.

- Conditions of incubation and hatching are not given.

-The procedures of the experiment are very vaguely described. Please describe the group activities in detail.

-It is also not known what was the exact number of tested samples in each of the methods

- Reference genes are not given

- Descriptions of tables and figures should be more extensive with an explanation of the abbreviations and research groups used.

- Figure 4 is incomprehensible

- The discussion lacks its own thoughts and suggestions

Author Response

(The authors gave the same response as above.)

Reviewer 3 Report

This article is generally interesting and well written. However, before the publication some points need to be clarified.

Abstract, page 3 etc.  – From histological point of view there are only four kinds of tissues: epithelial, connective, muscular and nervous. Therefore, such term as “liver tissue” is not justified. The authors terribly confuse organs with tissues. Organs are assembled from the four basic types of tissues and have cells with specialized functions.

Page 2 – please describe your hypothesis properly.

Page 2 – please explain what TNF-α, IL-1β, and IL-6, stand for.

Page 3 – please describe the method of animal euthanasia.

Page 11-12 – the authors should use italics if they refer to genes’ names

Figure 1 – scale bars are missing. In general, the micrographs are too small to make any judgement.

Table 4 – please rearrange the table. Gene symbols should be listed in one column.

Conclusions - In the present form the conclusion has no sign of novelty. The role of TLR/MAPK signaling in LPS-induced liver injury has been already described (see Chen Y, Guan W, Zhang N, Wang Y, Tian Y, Sun H, Li X, Wang Y, Liu J. Lactobacillus plantarum Lp2 improved LPS-induced liver injury through the TLR-4/MAPK/NFκB and Nrf2-HO-1/CYP2E1 pathways in mice. Food Nutr Res. 2022 Jul 5;66. doi: 10.29219/fnr.v66.5459.) This must be corrected. Some future perspectives are also needed.

Author Response

(The authors gave the same response as above.)

Round 2

Reviewer 3 Report

The authors responded to my comments in a reasonable manner.